# Application of Deep Learning Algorithm to Monitor Upper Extremity Task Practice

**DOI:** 10.3390/s23136110

**Published:** 2023-07-03

**Authors:** Mingqi Li, Gabrielle Scronce, Christian Finetto, Kristen Coupland, Matthew Zhong, Melanie E. Lambert, Adam Baker, Feng Luo, Na Jin Seo

**Affiliations:** 1Department of Computer Science, School of Computing, Clemson University, Clemson, SC 29634, USA; mingqil@clemson.edu (M.L.); mzhong41@gatech.edu (M.Z.); melambe@g.clemson.edu (M.E.L.); luofeng@clemson.edu (F.L.); 2Department of Health Sciences and Research, College of Health Professions, Medical University of South Carolina, Charleston, SC 29425, USA; scronce@musc.edu (G.S.); finetto@musc.edu (C.F.); coupland@musc.edu (K.C.); bakerdon@musc.edu (A.B.); 3Ralph H. Johnson VA Health Care System, Charleston, SC 29401, USA; 4Summer Intern, Research Experience for Undergraduates, Georgia Institute of Technology, Atlanta, GA 30332, USA; 5Department of Rehabilitation Sciences, College of Health Professions, Medical University of South Carolina, Charleston, SC 29425, USA

**Keywords:** stroke, upper extremity, rehabilitation, accelerometer, inertial measurement unit (IMU), wearable sensor, machine learning, deep learning

## Abstract

Upper extremity hemiplegia is a serious problem affecting the lives of many people post-stroke. Motor recovery requires high repetitions and quality of task-specific practice. Sufficient practice cannot be completed during therapy sessions, requiring patients to perform additional task practices at home on their own. Adherence to and quality of these home task practices are often limited, which is likely a factor reducing rehabilitation effectiveness post-stroke. However, home adherence is typically measured by self-reports that are known to be inconsistent with objective measurement. The objective of this study was to develop algorithms to enable the objective identification of task type and quality. Twenty neurotypical participants wore an IMU sensor on the wrist and performed four representative tasks in prescribed fashions that mimicked correct, compensatory, and incomplete movement qualities typically seen in stroke survivors. LSTM classifiers were trained to identify the task being performed and its movement quality. Our models achieved an accuracy of 90.8% for task identification and 84.9%, 81.1%, 58.4%, and 73.2% for movement quality classification for the four tasks for unseen participants. The results warrant further investigation to determine the classification performance for stroke survivors and if quantity and quality feedback from objective monitoring facilitates effective task practice at home, thereby improving motor recovery.

## 1. Introduction

Stroke is a leading cause of disability among adults in the United States, with approximately 800,000 people experiencing a stroke each year [1]. Upper extremity impairment is a common consequence of stroke, affecting 77% of people post-stroke [2,3,4]. Upper extremity impairment results in a decreased ability to perform functional tasks, negatively impacting individuals’ ability to perform activities necessary for self-care, hygiene, employment, and recreation, resulting in diminished independence and quality of life [5,6]. 

Research demonstrates that the recovery of upper extremity movement after stroke is enhanced with repeated task practice [6,7,8]. However, the extensive number of task repetitions necessary for recovery cannot be achieved through typical rehabilitation therapy visits [9,10,11]. To make up for this, rehabilitation therapists frequently supplement in-person treatment with the addition of a home exercise program (HEP) consisting of self-directed therapeutic activity [12]. Patient adherence to the HEP is typically assessed through self-report [13,14]. However, self-reporting adherence to the HEP is known to be inaccurate [13,15]. In addition to an inadequate measure of movement quantity, monitoring that cannot assess movement quality may lead to compensatory or inefficient task practice that does not restore function [16,17]. Varying degrees of HEP adherence at home may be a cause of heterogenous response to intervention among stroke survivors [12,13,15,18,19]. Furthermore, a lack of adherence at home may be responsible for the lack of transfer of improved motor capacity from therapy to the use of the affected upper extremity in daily living [10,19].

Remote monitoring of the HEP for both quantity and quality may help increase the use of the affected extremity in daily living. Remote monitoring may enable therapists’ review of patients’ HEP to proceed in an objective manner and enable the use of the patient’s clinic visit time to address barriers for HEP adherence or amend compensatory movement patterns to increase efficiency of therapy service. Thus, there is a crucial need for the technology and methodology to monitor movement quantity and quality in the home to provide objective measures of upper extremity task practice. 

The literature shows potential for such monitoring of upper limb movement using wrist-worn inertial measurement unit (IMU) sensors, as shown in Table 1. While a greater scope of IMU usage has been reviewed in a recent systematic review and meta analysis [20], Table 1 focuses on the previous work that used only wrist-worn IMUs for their usability and acceptability in the patient population of interest. Of these previous works, one study described the use of a wearable IMU sensor during exercise and was able to distinguish compensatory movements relative to correct (non-compensatory) movements [21]. However, they studied only a one-arm raise task in the sagittal and coronal planes, which is limited in the involvement of the distal upper extremity for object manipulation as relevant for activities of daily living as well as in the number of tasks typically administered for HEP. In addition, the previous study used 98 trials of this task from 11 stroke survivors, and it is possible that the samples may not have encompassed all varieties of compensatory movements following stroke. Further, the performance of deep learning methods has not been examined for this application. 

The contribution of this study is to address these limitations in previous studies by examining multiple tasks involving object manipulation relevant for activities of daily living, including varieties of post-stroke compensatory and incomplete movements, and utilizing a deep learning approach for this purpose. Specifically, we aimed to explore if deep learning may be a viable method to classify multiple representative tasks and task practice qualities. As the first step, the objective of the present study was to obtain IMU data during the simulation of various movement patterns for multiple representative HEP tasks by neurotypical adults with experience in neurorehabilitation and investigate the performance of the deep learning approach for the classification of the tasks and movement qualities. The anticipated impact of this work is a development of an objective monitoring system for HEP movement quantity and quality for stroke survivors, which will mitigate issues of self-reports and monitor movement patterns to achieve efficient rehabilitation service and optimal recovery. 

The materials and methods section describes the human participants of the study, the data collection procedure, and the deep learning classifier model selection, model evaluation, as well as data processing. In the results section, the classifier results for task identification and quality identification are presented. The discussion section summarizes the main findings, discusses the details of the results with task-specific interpretations and implications on future research design and approaches, and considers the future directions and potential impacts. 

## 2. Materials and Methods

### 2.1. Participants

Twenty neurotypical adults participated in this study. The mean age was 50.3 years (standard deviation of 18.4 years). The age range approximately matches the ages of stroke survivors that are typically seen for rehabilitation. All participants were experienced in assessment and treatment of motor impairment after stroke with clinical practice and/or research backgrounds, or as experienced caregivers of people with stroke. Ten were males and ten were females. They were free of orthopaedic or neurologic conditions limiting upper limb movement, compromised skin integrity of the wrist, or language barrier or cognitive impairment that precluded following instructions and/or providing consent. Written informed consent was obtained from all participants prior to their study participation.

### 2.2. Procedure for IMU Data Collection 

Participants came to the laboratory to perform 4 upper extremity tasks that are representative of tasks prescribed in HEP, while wearing an IMU sensor on their wrist (ActiGraph GT9X link, Actigraph Corporation, Pensacola, FL, USA). The IMU sensor recorded acceleration, gyroscope, and magnetometer data in 3 dimensions at 100 Hz. The tested side (right versus left hand) alternated between participants. Of the 20 participants who completed the study, 19 were right-handed with 10 using the right hand to perform the tasks and 9 using the left hand to perform the tasks. One participant was left-handed and used the right hand to perform the tasks. The 4 tasks were as follows: bring a cup to a shelf, bring a cup to the mouth, use tongs to transport an object, and move an object from one location to another (Table 2). Participants were instructed to perform the tasks with the specific conditions listed in Table 3. The list of conditions was compiled by two experienced occupational and physical therapists based on their clinical expertise in treating upper extremity movements after stroke. The conditions were organized under the umbrella of correct, compensatory, and incomplete movement qualities. The idea behind these three movement qualities is that if a person has many incomplete movements, it may suggest that the prescribed home tasks may be too difficult for the person and an adjustment of the task difficulty may be needed [28], while use of compensatory movements may indicate a need to practice the correct movement pattern to break out of the abnormal synergy patterns post stroke [29]. Participants were instructed to repeat each condition 10 times. For compensatory and incomplete movements, participants were asked to mimic movement patterns of people with stroke. 

### 2.3. Description of Classifiers

Two classifications were determined. The first was to identify which of the four tasks was being performed. The second was to classify the movement quality. The first classification was deemed relevant for future application in case the person performs the wrong tasks, which occurs frequently in the manual HEP. The second classification was deemed relevant for future application in order to monitor the quality of task practices, use this as feedback to improve movement quality, and trigger a consultation with a clinician. For quality classification, we considered two scenarios: (1) classify for movement conditions listed in Table 3 and then combine the results for the three movement qualities; (2) classify directly for the correct, compensatory, and incomplete movement qualities. 

### 2.4. Model Selection

Both classifications were based on the time series IMU data. Traditional machine learning methods like decision trees, random forests, and support vector machines rely on statistical features. These methods often necessitate manual feature engineering and struggle to effectively capture the sequential patterns inherent in the data. recurrent neural networks (RNN) are a logical choice for this type of data since they are well-suited for sequential data on classification tasks [30]. The input of the RNN cell consists of previous hidden states and the current input. Therefore, RNNs can maintain memories of previous states of the input data, and thus generate representations of the current state. In the context of our tasks, movements are dependent on the relation between the current state of the sensor and its previous state. Thus, those potential relationships from the data could be extracted to identify task IDs and conditions/qualities automatically using deep learning models.

However, vanilla RNNs can be challenging when the input sequence is long, and it might encounter the vanishing gradient problem because the gradients may approach zero after many time steps and cannot update weights effectively. Several variants of RNNs are proposed to address this problem and improve the performance. Long short-term memory (LSTM) [31] was introduced to handle the vanishing gradient problem and is capable of learning long-term dependencies. LSTMs use three different gates (input gate, output gate and forget gate) to add or remove information. Gated recurrent units [32] are another variant of RNNs that incorporate an update gate and a reset gate. GRUs are a widely utilized network because of their efficiency and simplicity. We conducted experiments on different RNNs and their variants. Based on the experimental results, we ultimately employed a bidirectional 2-layer LSTM with 512 hidden units and a dropout layer for all subsequent experiments. We optimized the model by Adam [33] with a 5e-4 weight decay. We set a batch size of 64. We used 512 as the maximum length of samples, since most of the data fell in this range and it was the best fit for our data and model. It took 20 h for 200 epochs on a 32GB NVIDIA V100 GPU. 

As for the movement quality classification, directly training for quality classification is challenging because of the class imbalance issue. The first classifier involved training with an equal number of samples for each condition. For the quality classifier, there are more conditions and thus more samples for the compensatory and incomplete qualities than for the correct quality, as seen in Table 3. The issue of class imbalance and the small amount of training data for the correct quality may impact the performance of the classifier. Thus, we experimented with over sampling and using weighted cross-entropy loss. The final model selection was based on the performance of the validation dataset as described below.

### 2.5. Model Evaluation

We evaluated the performances of the classifications in two ways: (1) classification of new movements from people whose data have been used for training the model (i.e., validation set), and (2) classification of movement from new people whose data have not been used for training the model (i.e., testing set). For the first method, we divided each participant’s data into the training set and validation set for each movement condition. Specifically, we randomly selected 8 out 10 samples for each movement condition for each participant as training data and the remaining 2 samples as validation data. The performance on the validation set indicates the model’s ability to classify new samples from the same participants whose information was used to train the model.

The second method involves the testing set to examine the model’s performance across different participants, using leave-one-person-out cross-validation. Specifically, we followed the same division mentioned above, but left one participant’s data out for testing. The model performance for the new participant’s data for which the model has not been trained was assessed. We repeated this step for each participant and combined the results to obtain an estimate of the model’s performance for new participants (i.e., testing dataset). This approach is commonly used to assess overfitting [34]. If the model performs too well on training data, it will simply memorize data rather than make an inference, and thus cannot generalize well on unseen data. 

### 2.6. Data Processing

Our dataset included the participant ID, task ID, quality label, and condition label. The IMU data file for each repetition of each condition was treated as a sample that consists of 12 features that depict the sensor’s movement over time. Specifically, the 12 features are the acceleration without gravity, the gravity vector, gyroscope readings, and magnetometer readings in 3 dimensions. The gravity vector was identified by estimating the sensor orientation from the accelerometer and gyroscope data using a six-axis Kalman filter (MATLAB 9.14.0.2206163 imufilter function). 

Consequently, each sample was represented as a 2D tensor whose shape is given in time series * feature dimensions. We processed the data to fit the time series’ length by either padding the data with zeros at the end if the sample length was shorter or down-sampling if the sample length was longer than the time series length. Additionally, we used RobustScaler from the Scikit-learn 1.3.0 package [35] to standardize our data to make sure all features are on the standardized magnitude. As a result, we transferred the IMU data to feature tensors that could be input into deep learning models.

## 3. Results

This section shows the classification results for task identification and quality identification. Results for both the validation dataset and the testing dataset from the leave-one-person-out cross-validation are provided. Both the accuracy and confusion matrices are provided to assess the model’s performance. 

### 3.1. Task Identification

Our model achieved an accuracy of 98.3% for the validation set, and 90.8% for the testing set. The confusion matrices for these two results are separately shown in Figure 1A,B. The results show that our model achieved impressive performance on data from participants whose other movement data were used for training. In other words, if we acquire a participant’s movement data and used it for the training process, our model had a high confidence in predicting the task ID of unseen samples from the same participant. Additionally, our model performed well even on unseen participants (for the testing set). 

### 3.2. Quality Identification

#### 3.2.1. Quality Identification via Classification of Movement Conditions

The results of the multi-class classifier for the movement conditions are shown in Figure 2. The results are shown for the validation set and the testing set separately for each task. These results for each condition were combined for the three movement qualities of correct, compensatory, and incomplete, as shown in Figure 3. This classifier achieved average accuracies of 91.7% for the validation set, and 74.4% for the testing set. The accuracy, precision, recall, and F1-score for each of the four tasks for both the validation and testing sets are presented at the top of Table 4. 

#### 3.2.2. Quality Identification via Direct Classification of Movement Qualities

The results of the multi-class classifier for the movement qualities are shown in Figure 4. The average accuracy reached 88.8% for the validation set. For the testing set, the average accuracy was 72.8%. The accuracy, precision, recall, and F1-score for each of the four tasks for both the validation and testing sets are presented at the bottom of Table 4. 

## 4. Discussion

This study examined if deep learning LSTM models can use the data from an IMU sensor worn on the wrist during upper extremity motor tasks to classify movement tasks and movement conditions/qualities. The results show that the model could classify the task being performed out of the four tasks with a 98.3% within-participant accuracy and a 90.8% between-participant accuracy. The movement qualities were classified with 94.8%, 93.6%, 87.4%, and 91.1% within-participant accuracy and 84.9%, 81.1%, 58.4%, and 73.2% between-participant accuracy for the four tasks, respectively, when specific movement conditions were initially classified and then combined for the three correct, compensatory, and incomplete movement qualities. When the three movement qualities were directly classified, 93.9%, 88.6%, 85.6%, and 87.2% within-participant accuracy and 80.8%, 76.5%, 63.0%, and 70.9% between-participant accuracy for the four tasks were obtained, respectively. With an improved accuracy compared to previous models using traditional machine learning approaches [21], the present study shows that deep learning models have the potential to aid in the monitoring of task practices. This study encourages future research with stroke survivors to investigate the utility of the monitoring of at-home task practices using the wearable IMU sensor and modeling to assist with their recovery process. 

Our classifier achieved impressive results for task identification. It performed well on unseen data from the same participants and generalized effectively to new participants. This high performance is likely because the four tasks are clearly distinct and between-participant variability is relatively minor. The results suggest that the data distribution was similar between healthy participants and our model could learn the underlying pattern.

For quality classification, the accuracy was higher for the validation set compared to the testing set. This result suggests that while classification was possible for unseen participants, it may be better to leverage some data from participants to the model if it is practically feasible. We used the leave-one-person-out classification in which we used 19 participants’ data for training (80% of data) and validation (20% of data) and one participant’s data for testing. During the training process, the classifier never saw the data used for testing. For validation, the classifier never saw the validation data, but it was trained using the data from the same participants. When the one test participant was different in terms of their limb size or movement speed or performed a task with a new movement pattern that the 19 other participants did not use, the classifier was expected to perform worse. This phenomenon may have contributed to the quality classification performance results. This indicates that if we can collect data from a person and use their own data during the training process, we can refine the performance of the classifier, further improving upon previous models. 

The overall quality classification was better when individual conditions were classified and lumped for the three movement qualities (correct, compensatory, and incomplete), compared to when the three-class classifier was used. The class imbalance issue may have impacted the performance of the three-class classifier, even though the weighted loss function was used. The condition classification itself could be used to help monitor the details of the compensatory and incomplete movements, and thus guide individuals to avoid the specific movement condition to improve their movement quality. However, feedback that is too detailed might not be as effective as allowing people to troubleshoot and explore their movement patterns by themselves, and thus feedback for task practices should be carefully designed [36,37]. 

The condition classification results provide insights into how the model performed with each movement condition. Some conditions could be classified effectively, while other conditions were not. Some conditions could be predicted effectively, while some conditions tended to be misclassified. For example, the results show that some of the movement conditions within the same quality were similar with each other and thus were difficult to distinguish from the IMU data. Specifically, in Task 1, incomplete conditions #1–4 all involved not being able to grasp the cup. Thus, these conditions were difficult to distinguish among themselves. Similarly, in Task 2, incomplete conditions #5–8 all involved not being able to lift the cup to different extents, and thus their classifications were mixed among themselves. Similar results were seen for the compensatory movement quality, such as shoulder elevation vs. shoulder retraction in Task 2. Thus, some of the conditions within each movement quality examined in this study may have been redundant for the purpose of classifying the movement quality. 

Our classifiers were able to fit the data from participants seen during the training process but tended to misclassify correct movements as compensatory for unseen participants. To further assess the classifiers, the condition classification results can be reviewed to identify which condition was likely misclassified for the movement quality. For example, the compensatory condition of using forward neck flexion to bring the cup to the mouth for Task 2 was classified as correct 5% and 19% of the time for within-subject and between-subject data (for validation and testing set), respectively. The neck flexion was not captured by the IMU sensor worn on the wrist, thus contributing to the misclassification for this condition. Similarly, for Task 4, distinguishing the different grasp types such as the key grip and the whole-hand grip from the pincer grip or the three-jaw chuck grip was difficult, likely because the IMU sensor worn on the wrist is minimally impacted by finger posture. Therefore, some of the compensatory conditions involving joints not directly measured by the sensor may be difficult to classify especially for new people. 

The classification performance varied across the four tasks. The classification accuracy was the highest for Task 1 and lowest for Task 3, likely due to different magnitudes and locations of joint motions occurring for each task. With the IMU sensor placed on the wrist, movement classification improved when the wrist was moved by a greater magnitude, such as when it was lifted to the top of the box for Task 1. The classifier for Task 3 performed decent on the validation set, while it obtained poor performance on the testing set. Task 3 involved manipulating two objects (the tongs and the block), while the other three tasks involved manipulating only one object. Further, the between-subject variability may have been greater for Task 3 due to differences in the hand size and strength contributing to where the subjects grasped on the tongs and thus how much they moved the wrist toward the targets. In addition, the travel distance for the wrist required to complete this task was smaller due to the use of the tongs, compared to the other tasks. As the task was focused on transferring the block rather than moving the tongs in prescribed conditions, various angular profiles of the tongs and the wrist could be used to accomplish the task, which may have hampered the detection of compensatory shoulder and trunk movements. The smaller travel distance may also explain misclassification for incomplete conditions such as not transporting the block all the way to the target. Thus, the task complexity, between-subject differences, and smaller movement distance altogether may have contributed to the classification performance of this task. 

Future work may consider the poor classification results of these conditions in the design of the application of the IMU-based task-practice monitoring system. Future directions include adapting our models to data from stroke survivors. In doing so, transfer learning may be applicable. Transfer learning has shown great success in deep learning applications in different domains or datasets [38,39,40,41,42]. Based on this, future work may utilize the model trained using healthy data and fine-tune the model for a small amount of data from stroke survivors. This approach may not be optimal compared to training the model with plenty of data from stroke survivors, but it is worth investigating if it could achieve acceptable performance without overburdening stroke survivors. Future work may also investigate model performance in case a person uses multiple compensatory movements simultaneously or employs unexpected movement patterns beyond what were examined in this study that may occur in real world applications.

The implication of this line of research is developing a monitoring system for the HEP for stroke survivors. Task identification and condition/quality classification can provide the basis for the objective monitoring of the HEP in quantity and quality. This monitoring can provide an opportunity for therapists and stroke survivors to review the HEP adherence level. This review can lead to conversations about barriers and facilitators for HEP adherence and address various psychosocial, socioeconomic, and personal factors relevant for HEP adherence and recovery. The monitoring of HEP movement quality can reveal compensatory movement patterns that patients use but need to try to reduce during HEP to achieve optimal recovery [43]. Feedback of the detailed condition detection for each HEP repetition can provide metrics for patients and therapists to work toward as a goal. In addition, feedback of a large number of incomplete tasks may signal a possibility that the prescribed HEP task is too difficult for the patient to complete and may need to be adjusted to achieve an optimal challenge level desired for recovery [44,45,46]. Ultimately, this HEP monitoring system is expected to enable progress- and adherence-driven visits for the increased efficiency of rehabilitation services. 

## 5. Conclusions

We show the potential of monitoring the tasks and movement qualities of upper extremity motor tasks using wrist-worn IMU sensor data and deep learning LSTM models. The results encourage future research to examine the classification performance for stroke survivors. Future direction includes investigation of the utility of objective monitoring and feedback for quantity and quality of at-home task practices to assist with motor recovery in stroke survivors. 

## Figures and Tables

**Figure 1 sensors-23-06110-f001:**
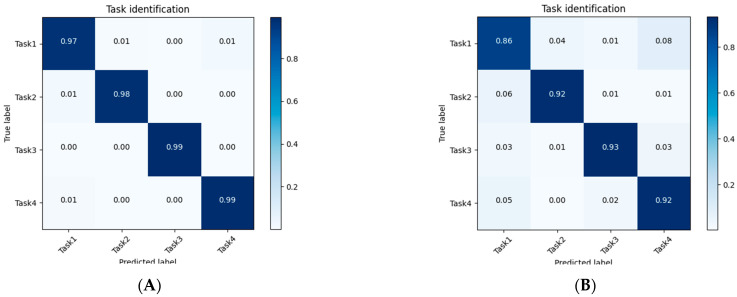
Task identification confusion matrices for results on the validation set (**A**) and the testing set from the leave-one-person-out cross-validation (**B**).

**Figure 2 sensors-23-06110-f002:**
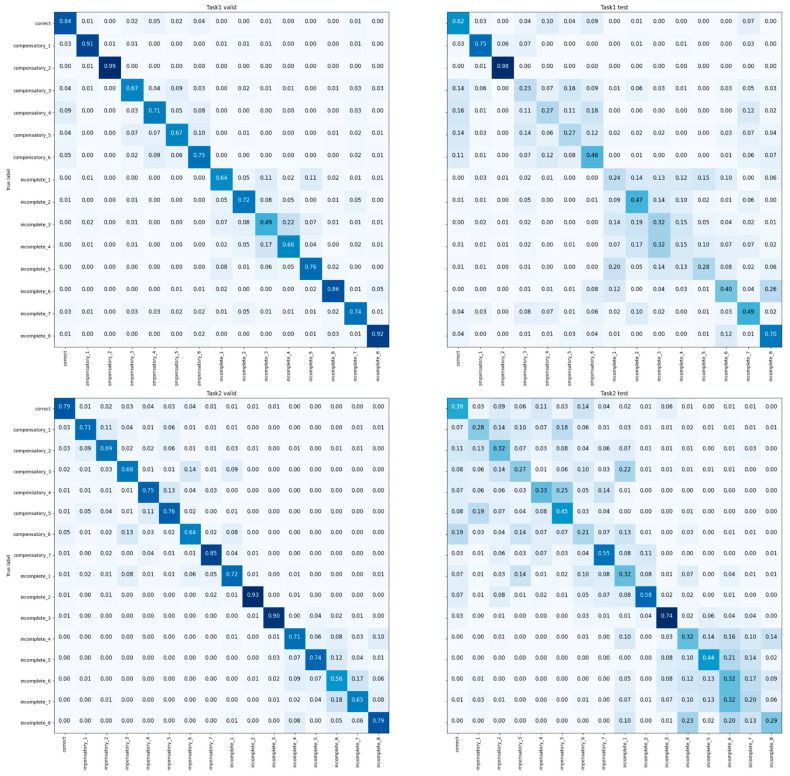
Condition classification confusion matrix on four tasks.

**Figure 3 sensors-23-06110-f003:**
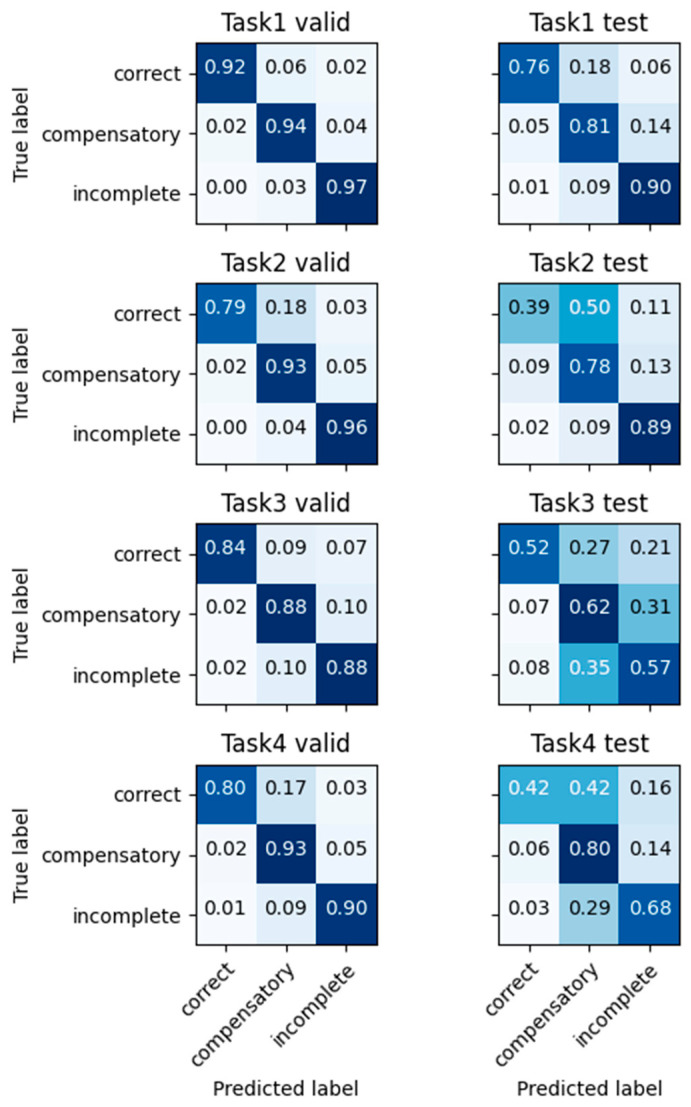
Condition classification results combined for each of the three movement qualities. Results of each task are shown from the top to bottom. Results for the validation set are on the left column. Results for the testing set are on the right column.

**Figure 4 sensors-23-06110-f004:**
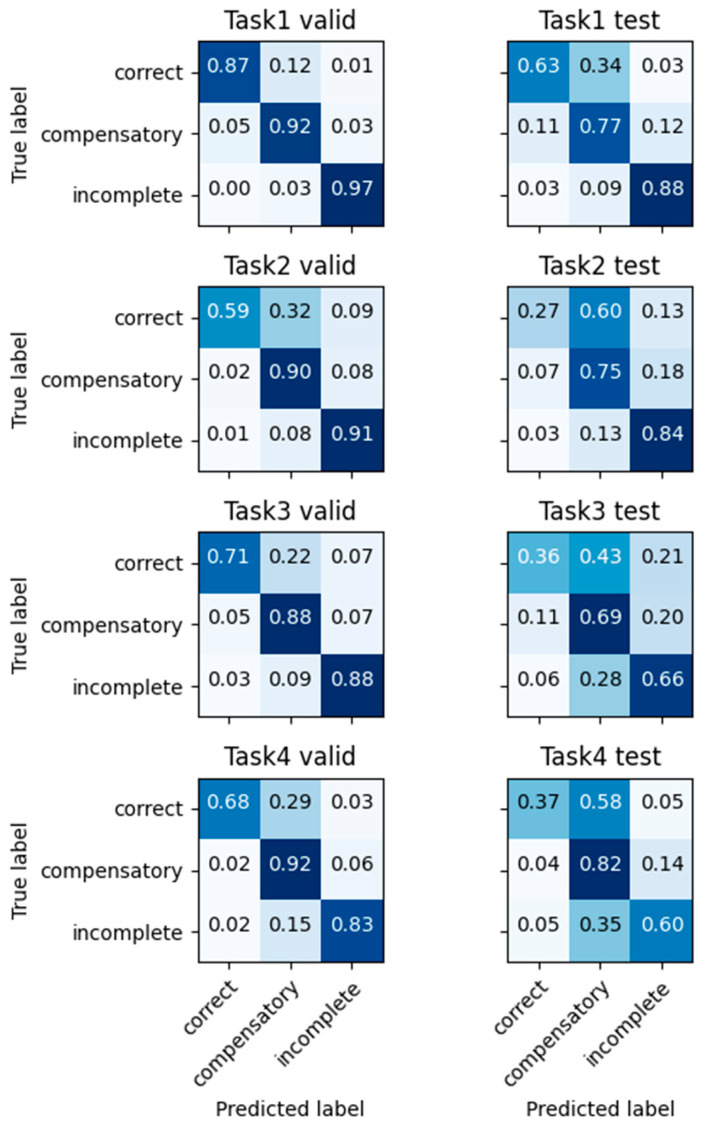
Quality classification confusion matrix on four tasks which are directly trained using quality labels. Results of each task are shown from the top to bottom. Results for the validation set are on the left column. Results for the testing set are on the right column.

**Table 1 sensors-23-06110-t001:** Summary of previous research classifying upper limb movement types using data from wrist-worn IMU sensors (NR = not reported, DTW = dynamic time warping, LSTM = long short-term memory, RF = random forest, CNN = convolutional neural network, GBM = gradient boosting machine, ET = extremely randomized trees, PD = Parkinson’s disease).

Reference	Goal	Participant	Machine Learning Approach	Cross Validation	Accuracy	Precision	Recall	F1
Bhagat 2020 [22]	Classify cylindrical vs. pincer grasp (to pick up a water bottle vs. pen) from reaching motion	2 persons with spinal cord injury	DTW	5-fold intrasubject	84.5%	NR	NR	NR
			LSTM	5-fold intrasubject	99%	NR	NR	NR
Gomez-Arrunategui 2022 [23]	Detect reach time during prescribed tasks	12 stroke survivors	RF	5-fold intrasubject	74.8%	58.8%	46.9%	NR
			CNN	5-fold intrasubject	76.5%	62.9%	43.0%	NR
Van den Tillaar 2021 [24]	Classify handball throw types (circular/whip, standing/running/jumping)	17 handball players	GBM	leave-one-person-out	83%	NR	80%	80%
Pfister 2020 [25]	Classify motor state (off, on, dyskinetic) in free-living	30 persons with PD	CNN	leave-one-person-out	65.4%	NR	65%	NR
Lee 2018 [21]	Classify quality of arm raise (healthy control, good, feedback needed)	9 healthy, 11 stroke survivors	RF	leave-one-person-out	82%	64.8%	65.2%	63.3%
Villalobos 2022 [26]	Classify musculoskeletal disorder risk level during meat cutting	20 meat cutters	ET	NR	97%	98%	96%	97%
Bochniewicz 2017 [27]	Classify functional vs. nonfunctional movement	10 healthy persons	RF	leave-one-person-out	91.53%	NR	NR	NR
		10 stroke survivors	RF	leave-one-person-out	70.18%	NR	NR	NR

**Table 2 sensors-23-06110-t002:** Description of the four tasks.

Task #	Task Name	Task Instruction	Task Photo
1	Cup to shelf	Start with the hand in the start position and the cup in the pre-set target. Use the cylindrical grasp to grasp the cup and move it to the top of the box. Extend the fingers to release the cup. Return the hand to the start position.	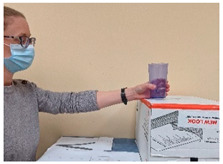
2	Cup to mouth	Start with the hand in the start position and the cup in the pre-set target. Use the cylindrical grasp to grasp the cup, raise it to approximately 1 inch from the mouth, return the cup to the target, and return the hand to the start position.	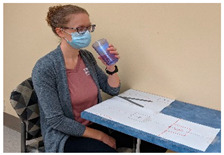
3	Tong use	Start with the hand lateral to the tongs, the tongs in the pre-set start position marked by “V”, and the block in target #1. Grasp the tongs, pick up the block on target #1 (near midline), move and release the block to target #2 (lateral), return the tongs to the start position, and return the hand to the start position.	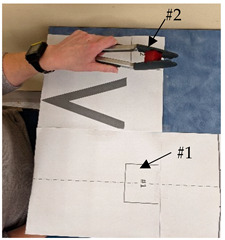
4	Finger food	Start with the hand in the start position and the block in target #3. Use the pincer or 3-jaw chuck grasp to move the block from target #3 to target #1 (farther from the participant to closer to the participant), release the block, and return the hand to the start position.	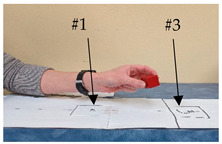

**Table 3 sensors-23-06110-t003:** Correct, compensatory, and incomplete movement qualities and conditions for each task.

Task	Movement Qualities	Conditions
1. Cup to shelf	Correct	Use normal movement patterns.
	Compensatory	Non-cylindrical grasp (e.g., from the top of the cup); Pronation of forearm to grasp; Shoulder and scapular elevation/shoulder hiking with lift of cup during reach to shelf; Forward flexion of trunk with lift of cup to reach shelf; Trunk lateral lean (away from tested side) with lift of cup to reach shelf; Task performed with slowed speed.
	Incomplete	Cannot grasp due to weakness; Too forceful of grasp yields inability to grasp cup; Dysmetria (alternating too far/too close) yields inability to grasp cup; Dysmetria (toward affected side) yields inability to grasp cup; Grasp completed but unable to lift cup from table; Grasped, cup lifted from table, unable to reach to shelf; Grasped, cup lifted from table, drops cup before reaching shelf; Grasped, lifted and reached top of shelf, unable to release grip from cup, bring cup back to start position and release using non-tested hand.
2. Cup to mouth	Correct	Use normal movement patterns.
	Compensatory	Shoulder and scapular elevation/shoulder hike; Scapular/shoulder retraction with lift of cup (exaggerated); Forward flexion of trunk with lift of cup; Trunk extension with lift of cup; Trunk lateral lean (away from affected side) with lift of cup; Forward neck flexion when lifting cup to mouth; Task performed with slow speed.
	Incomplete	Cannot grasp due to weakness; Too forceful of grasp yields inability to grasp cup; Dysmetria (alternating too far/too close) yields inability to grasp cup; Dysmetria (toward affected side) yields inability to grasp cup; Grasp completed but unable to lift cup from table; Grasp completed, cup lifted from table, unable to reach mouth; Grasped, lifted to mouth, unable to release grip from cup, non-tested hand used to release cup; Grasp completed, cup lifted from table, drops cup before reaching mouth.
3. Tong use	Correct	Use normal movement patterns.
	Compensatory	Shoulder/scapular elevation/shoulder hiking with transfer of block; Excessive scapular/shoulder retraction with lift/transfer of block; Forward trunk flexion with reach/lift of tongs/transfer of block; Trunk lateral lean (away from tested side) with reach/lift/transfer of block; Task performed with slowed speed; Sliding arm or hand across table to reach tongs, block, or targets.
	Incomplete	Cannot grasp tongs due to weakness; Dysmetria (too close) yields inability to pick up block; Dysmetria (toward affected side) yields inability to pick up block; Grasped tongs, picked up block, unable to reach target #2, tongs and block returned to target #1; Grasped tongs, lifted block and reached to target #2, unable to release block, block returned to target #1 and removed from tongs with non-tested hand; Grasp completed, tongs lifted from table, drops block before reaching target.
4. Finger food	Correct	Use normal movement patterns.
	Compensatory	Key grip used; Raking/full-handed grasp used; Sliding block off table to lift; Scapular elevation/shoulder hiking with lift of block or reach; Excessive scapular/shoulder retraction with lift/transfer of block; Forward flexion of trunk with reach/lift/transfer of block; Trunk extension with lift/transfer of block; Trunk lateral lean (away from tested side) with reach/lift/transfer of block; Task performed with slowed speed; Slide arm across table to reach block/transport block.
	Incomplete	Cannot grasp due to weakness; Too forceful of grasp yields inability to grasp block; Dysmetria (too close) yields inability to grasp block; Dysmetria (toward affected side) yields inability to grasp block; Unable to extend arm to reach target #3; Unable to release grip on block once moved to target #1, remove block with non-tested side; Drops block before reaching target #1.

**Table 4 sensors-23-06110-t004:** Accuracy, precision, recall, and F1-score for four tasks using the condition and quality classification.

		Validation Set (Within-Participant)				Test Set (Leave-One-Person-Out)			
	Task #	Accuracy	Precision	Recall	F1-Score	Accuracy	Precision	Recall	F1-Score
Using condition classification	Task 1	94.8%	93.4%	94.3%	93.9%	84.9%	80.0%	82.3%	81.0%
	Task 2	93.6%	91.0%	89.3%	90.2%	81.1%	67.4%	68.7%	67.9%
	Task 3	87.4%	85.0%	86.7%	85.8%	58.4%	53.3%	57.0%	54.7%
	Task 4	91.1%	86.4%	87.7%	87.0%	73.2%	62.0%	63.3%	62.5%
Using quality classification	Task 1	93.9%	88.6%	92.0%	90.1%	80.8%	71.0%	76.0%	72.7%
	Task 2	88.6%	83.8%	80.0%	81.7%	76.5%	62.0%	62.0%	62.0%
	Task 3	85.6%	79.5%	82.3%	80.8%	63.0%	55.4%	57.0%	55.8%
	Task 4	87.2%	81.4%	81.0%	81.1%	70.9%	60.1%	59.7%	59.5%

## Data Availability

Data will be shared upon request.

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
