# Peer review of "Application of Deep Learning Algorithm to Monitor Upper Extremity Task Practice"

_sensors, 2023, doi:10.3390/s23136110_

Round 1

Reviewer 1 Report

The issue of helping to monitor hemiplegia exercises is very laudable. Congrats.

1) The title is too ambiguous: “Development of algorithms”?, I suggest "Application of (Classifier Name)" in the problem at hand

2) I suggest to the authors a representative sample of at least 30 patients to give greater validity.

3) I suggest that in the introduction section, the authors add a concise paragraph of the contributions of the article.

4) Also in the introduction section (at the end), add a paragraph to describe what will be seen in each paragraph

5) In the introduction section, add a table, which is compared with other articles to highlight the novelty of your proposal. In other words, what elements are not found in other jobs than in this one? A brief example can be found in Tables 1 and 2 of this article: https://www.mdpi.com/2227-7390/11/11/2524

6) Take care of the table format, the same as MDPI suggests.

7) If authors use LSTM NN, then they must justify the selection of this approach and not other classifiers, such as KNN, Decision Tree among others.

8) For your model validation, by leaving a patient out of the training and using it for testing, I think it is a leave-one-out cross validation, if so, comment on it, if not, explain it better because I don't see the point of using a 20 then -fold.

9) As it is a classification problem, authors only used the accuracy metric, but they should strengthen their results with other metrics, such as precision, recall, f1-measure, since each one measures different things. These metrics are derived precisely from the confusion matrix. These metrics are necessary to validate your method.

Reviewer 2 Report

The paper is interesting and generally well written, however some points should be addressed.

- while the literature review focuses on stroke, there is very little (none actually) of task identification/classification from wrist-worn IMU sensors under other applications (ie not necessarily stroke-related).  This is highly relevant and correlated, and needs to be addressed. The paper gives no sense of the state of the art with respect to task classification from wrist worn sensors.

- Please give details technical specifications and the reason for choosing the specific IMU sensor chosen. Ie. why choose the ActiGraph GT9X link? Presumably, for actual practical implementation of this approach a 'regular' smartwatch would be the hardware of choice. So how do the specifications of the actigraph compare against, for example, an apple watch? how do the specifications of the chosen IMU sensor affect the results? can we expect comparable results from an apple/garmin smartwatch? 

- the explanation of what constitutes 'validation set' and 'testing set' is not clear. All the results that follow are based on the reader understanding what is the distinction between the two. I think I inferred that ' validation set' referred to new data/seen person (new data from same person whose data was used to train) and 'training set' was new data/unseen person(new data from a person whose data was not used to train).  Am I correct? why am I doubting this? why not make it more clear to the reader so that there are no doubts?

-Typos. Line 293 "even though weighted lost function" --- I think "lost" should be "loss".

- Typos. Line 307 "in different extents" should be "to different extents".

Round 2

Reviewer 1 Report

Thank you for your response